# Enhancing Bitumen Properties through the Utilization of Waste Polyethylene Terephthalate and Tyre Rubber

Omar R. Khaleel [1], Laila K. N. Al Gharbi [2] and Moatasem M. Fayyadh [3,*]

1. Civil Engineering, Engineering College, Al-Iraqia University, Baghdad 10011, Iraq
2. Civil Engineering, Middle East College, Knowledge Oasis, Muscat 112, Oman
3. Asset Lifecycle, Sydney Water, Parramatta, NSW 2150, Australia
* Correspondence: moatasem.m.f@gmail.com

**Abstract:** The disposal of waste accumulation has become a significant challenge in Oman due to the increasing population. Co-biodegradability issues arise from accumulating two types of non-co-biodegradable waste materials: plastic and rubber. Asphalted pavements experience various stresses resulting from high traffic density, leading to numerous problems. This study aims to investigate the impact of incorporating waste plastic and rubber on the engineering properties of bitumen. Specifically, the study examines the addition of waste tyre rubber (4% and 6%) and waste polyethylene terephthalate (PET) (4% and 6%) to bitumen with a grade of 85/100. Three tests were conducted to evaluate the physical properties of the bitumen, including softening points, penetration tests, and viscosity tests. The results demonstrate that the penetration of bitumen with the addition of 6% waste tyre rubber and PET was 9% lower compared to the 4% mixture for both waste materials. Furthermore, tyre rubber-modified bitumen exhibited higher softening points (79 °C, 2580 s) and viscosity when compared to plain bitumen (48 °C, 1800 s) and PET-modified bitumen (53 °C, 2150 s). These differences indicate that incorporating waste PET and tyre rubber improves the engineering properties of bitumen. This study highlights the importance of increasing the softening point of bitumen in Oman's high-temperature areas.

**Keywords:** PET; tyre rubber; bitumen; engineering properties





## 1. Introduction

Technological advances, widespread industrialization, and consumer habits have intensified the accumulation of unmanageable waste. Among the various waste products, plastic and rubber pose significant challenges as they are non-biodegradable and detrimental to the environment. Proper disposal of plastic and tyre rubber is crucial, particularly in urban and rural areas, where these materials can be recycled and utilized to enhance the properties of bitumen [1]. In Oman, the annual accumulation of these critical waste categories continues to increase, posing a pressing concern.

Asphalt pavement is subjected to multiple stressors, including fatigue, ravelling, and rutting, mainly due to high traffic density, vehicle weight, and speed. Various modifications have been explored to improve the performance of asphalt binders [2,3]. Numerous studies have focused on enhancing the properties of bituminous materials used as binders by incorporating different waste materials [3,4].

Airey et al. [5] reported that using tyre rubber in asphalt mixtures reduces fatigue cracks, enhances tensile strength and toughness, improves skid resistance, and increases elasticity. Zolfaghari et al. [6] investigated the effect of different sizes of tyre rubber crumbs on bitumen and concluded that reducing the size of crumb rubber enhances the penetration value of bitumen. Similarly, Sernas et al. [7] examined the impact of rubber content on bitumen properties using different rubber content percentages and bitumen grades. Their findings demonstrated that an increase in rubber content improves penetration

resistance. Mashaan et al. [8] found that incorporating crumb rubber into bituminous materials enhances the softening points of bitumen. Their study revealed that increasing the blending temperature increases the mass of tyre rubber due to rubber swelling during the mixing process. Furthermore, the study highlighted that increasing rubber content effectively improves the stiffness properties of bitumen.

Several investigations have indicated that adding tyre rubber significantly enhances the viscosity of bitumen [9–12]. Experiments have shown that the melting point of bitumen increases to 53 degrees Celsius with the addition of 1.5% plastic waste and further rises to 56 degrees Celsius with the addition of 6% plastic waste. Additionally, a higher percentage of plastic waste (6%) results in bitumen with higher viscosity (1600 cp), compared to 500 cp obtained with a lower percentage. Thamme Gowda and colleagues [13] concluded that adding thermoplastic materials to bitumen improves viscoelastic behavior. Singh and Ghauhan [14] found that adding 12% crumb rubber to bitumen produces favorable results, including an increased softening point and reduced penetration and ductility.

Studies have also investigated the impact of different plastic wastes, such as polypropylene, on the performance of construction materials. Mazloom and Mirzamohammadi [15] studied the thermal effect on the mechanical properties of mortars containing polypropylene fibres and observed an improvement in compressive strength. Touahri et al. [16] explored the effect of recycled polypropylene fibre on concrete. They found that it enhances mechanical performance, including higher flexural strength, reduces drying shrinkage, and increases permeable pore voids. Pamudji et al. [17] examined the effect of polypropylene waste as coarse aggregate on the behavior of reinforced concrete beams. They observed a noticeable reduction in concrete density and improved structural performance.

According to Taherkhani and Arshadi [18], transforming scrap tyres into crumb rubber involves mechanical grinding under varying conditions. The crumb rubber is then incorporated into asphalt mixtures using two distinct procedures: the wet and dry processes. In the wet process, rubber is blended with asphalt cement at high temperatures using specialized mixers, and the resulting modified asphalt is then combined with hot aggregate in the asphalt plant mixer. In addition, in the dry process, crumb rubber particles are added directly into the aggregate and mixed with hot asphalt cement in the asphalt plant mixer. A critical aspect affecting the performance of rubber-modified binders is crumb rubber and asphalt interaction. This interaction involves simultaneous mechanisms of swelling and dissolution. When rubber particles are added, the asphalt's aromatic oils are absorbed into the polymer chains of the crumb rubber, causing the rubber particles to swell up to two to three times their original volume and form a gel-like substance. Consequently, the swelling of rubber particles decreases the distances between them, leading to a tenfold increase in binder viscosity. Studies conducted on rubber-modified binders and mixtures have indicated significant improvements in properties such as rutting, fatigue, thermal, and reflective cracking resistance.

According to Mensahn et al. [19], using waste rubber particles in asphalt and asphalt mixtures enhances flexible pavement performance while adhering to environmental regulations. These waste rubber particles can be adjusted for use in open-graded, gap-graded, and dense-graded asphalt pavement, taking into account gradation limits. Additionally, the design mix for ensuring the performance stability of waste rubber particles involves determining the aggregate gradation type, the gradation and percentage of waste rubber particles, optimal bitumen content, and the type of asphalt pavement. Mensahn and colleagues further observed that the control mix design exhibits varying performance effects, considering the percentage dosages of waste rubber particles in the dry and wet processes. The Marshall Stability Test method is a comprehensive means of assessing the structural strength and deformation of the engineering properties of waste rubber particles. In regions with hot temperatures, using waste rubber particles in flexible road pavement can help reduce associated pavement issues such as thermal cracking, fatigue, and permanent deformation (rutting).

Jeong et al. [12] investigated the effect of crumb rubber on bitumen viscosity. They added two different percentages of tyre rubber, 10% and 20%, and determined that 20% of tyre rubber resulted in higher viscosity bitumen than 10%. Khalid and Artamendi [20] also indicated that the viscosity of bitumen increases with crumb rubber, especially at percentages higher than 10%. Neslihan Atasagun [21] demonstrated that combining PET with waste paper improves the viscosity and softening point of pure bitumen.

The primary objective of this study is to investigate the influence of waste plastic, specifically polyethylene terephthalate (PET), and waste tyre rubber on the engineering properties of bitumen. The research aims to address the challenges related to the softening point of bitumen in hot weather conditions and the occurrence of potholes in Oman. By examining the effects of these waste materials on the properties of bitumen, valuable insights can be gained to improve the performance and durability of asphalt pavements, ultimately contributing to sustainable waste management and infrastructure development.

## 2. Materials and Methods

### 2.1. Materials

2.1.1. Bitumen

Bitumen is a dark or black solid material obtained from the distillation of petroleum. It is a viscous adhesive substance consisting of high molecular weight hydrocarbons. At room temperature, bitumen is in a solid state; at 100 °C, it becomes highly viscous [22]. In this study, bitumen grade 85/100 was used. Table 1 presents the physical properties of bitumen grade 85/100, including specific gravity, penetration, and softening point.

**Table 1.** The properties of bitumen grade 85/100.

| Bitumen 85/100 | ASTM Standard Limits | Standard |
|---|---|---|
| Specific gravity at 25 °C | 1.01–1.05 | ASTM D-70 |
| Penetration at 25 °C, 5 s, 100 g, 0.1 mm | 85–100 | ASTM D-5 |
| Softening point (°C) | 45–52 | ASTM D-36 |

2.1.2. Tyre Rubber

A tyre is a pneumatic covering made of synthetic or natural rubber or a combination thereof that encloses a wheel. Tyres are categorized based on their specific usages, such as truck and car tyres [23]. In this study, the tyre rubber used was obtained from United Rubber Industries (URI), ranging in size from 1 mm to 3 mm. Table 2 illustrates the percentage compositions of tyre rubber for truck and car tyres. The primary components of tyres include additives, sulphur, zinc oxide, textile, metal, carbon black, and rubber [23].

**Table 2.** The chemical compositions of tyre.

| Ingredient | Percentage Composition—Truck | Percentage Composition—Car |
|---|---|---|
| Additives | 5% | 7.5% |
| Sulphur | 1% | 1% |
| Zinc oxide | 2% | 1% |
| Textile | - | 5.5% |

**Table 2.** *Cont.*

| Ingredient | Percentage Composition—Truck | Percentage Composition—Car |
|:---:|:---:|:---:|
| Metal | 25% | 16.5% |
| Carbon black | 22% | 21.5% |
| Rubber | 45% | 47% |

### 2.1.3. Plastic (PET)

PET, which stands for polyethylene terephthalate, is a long-chain polymer that falls under the group of polyesters. PET is formed through a polymerization reaction between alcohol and acid. It is known for being easy to handle, robust, and long-lasting [24]. In this study, the polyethylene terephthalate used was obtained from Octal Company, with a diameter of 2.5 mm. Table 3 presents the properties of polyethylene terephthalate [24].

**Table 3.** The properties of PET.

| Properties | Value | Standard |
|:---|:---:|:---:|
| Specific gravity | 1.28 | |
| Melting temperature | 265 °C | |
| Moisture absorption | 0.1 | ASTM D792 |
| Tensile strength | 850 kPa | |

### 2.2. Sample Preparation

Conventional Bitumen: A 500 g sample of normal bitumen was prepared in a small pan and heated in the oven until it reached a temperature of 160 °C. Subsequently, three tests were conducted on the bitumen.

Modified Bitumen: To prepare modified bitumen, approximately 500 g of bitumen was heated until it turned into a liquid state. The bitumen was then heated to 160 °C, adding 4% of tyre rubber. The modified bitumen was manually mixed and placed on the penetration apparatus.

As depicted in Figure 1, the three materials utilized in this study were bitumen, waste plastic PET, and waste tyre rubber. Two different percentages, 4% and 6%, of plastic PET and tyre rubber were added to 500 g of bitumen grade 85/100, as outlined in Table 4. The two-tube Saybolt apparatus was employed to evaluate the samples.

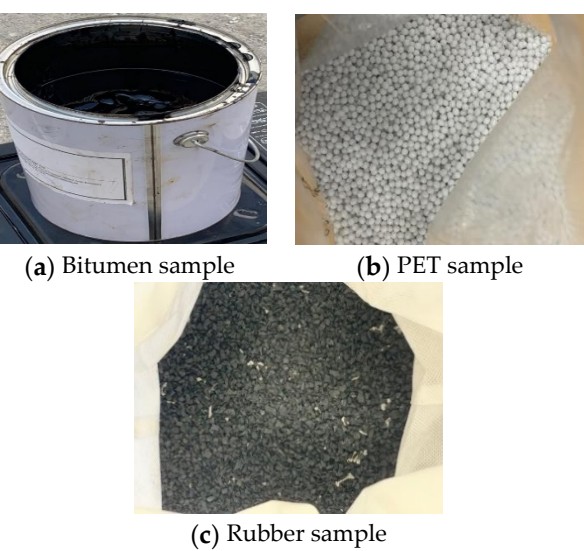

(**a**) Bitumen sample        (**b**) PET sample

(**c**) Rubber sample

**Figure 1.** Samples of the used materials.

**Table 4.** Sample details.

| Sample Number | Sample | Bitumen (gram) | Tyre Rubber (gram) | Polyethylene Terephthalate (PET) (gram) |
|---|---|---|---|---|
| 1 | Normal bitumen | 500 | - | - |
| 2 | Normal bitumen + 4% tyre rubber | 500 | 20 | - |
| 3 | Normal bitumen + 6% tyre rubber | 500 | 30 | - |
| 4 | Normal bitumen + 4% PET | 500 | - | 20 |
| 5 | Normal bitumen + 6% PET | 500 | - | 30 |

*2.3. The Experimental Tests*

The details of the adopted tests are shown below.

### 2.3.1. Penetration Test (ASTMD D-5 [25])

The Penetration Test is utilized to determine the consistency or hardness of the bitumen. To prepare the test specimen, the bitumen material was softened at 160 °C and then poured into a container, filling it to a depth of 10 mm. Subsequently, the sample was allowed to cool to the room temperature of 25 °C, and the container was positioned on the penetration apparatus. An initial reading was recorded after ensuring the needle made contact with the sample's surface. A total load of 100 g was applied, and the needle was released for 5 s. Eventually, a final reading was recorded.

### 2.3.2. Softening Point Test (ASTMD D-36 [26])

The Softening Point Test is conducted to determine the softening point of the bitumen. The test samples were prepared following the ASTM D-5 standard. Initially, the samples were softened at a temperature of 160 °C. The softened bitumen was then poured into rings placed in a frame, and this entire system was immersed in a water-filled bath at a temperature of 5 °C. Balls were placed in the rings and heat was applied to the bath. The temperature at which the ball made contact with the bottom surface of the plate was noted. The same procedure was repeated for the other ball, and the average softening point values were recorded as the softening point (ASTM D 2019).

### 2.3.3. Saybolt Two-Tube Viscometer Test (ASTM D-88 [27])

The Saybolt viscometer test is employed to measure the viscosity of the bitumen. The temperature of the bath was carefully regulated. The cork stopper was adequately inserted at the viscometer's lower part to prevent air from escaping. The sample was preheated and stirred before being dispensed. The receiving flask was positioned correctly, and upon removing the cork, the timer was initiated. The time was recorded from the start of the flow until the last drop.

## 3. Results and Discussion

The results of the penetration test on various samples are presented in Figure 2, illustrating the impact of different amounts of PET waste and waste tyre rubber on the resistance to water penetration of bitumen. Compared to conventional bitumen, the penetration values decreased by 15% and 23% when 4% and 6% of waste tyre rubber were used as additives, respectively, as indicated by the findings of Manoharan et al. [28]. These results indicate that as the percentages of additives increase, the penetration values decrease. This improvement can be attributed to the ability of waste tyre rubber to enhance the stiffness of bitumen. The interaction between the waste tyre rubber and hot bitumen

leads to an increase in the mass and volume of the rubber. As a result of this interaction, the rubber expands due to the absorption of the light oil fraction along with the bitumen [1,8,19].

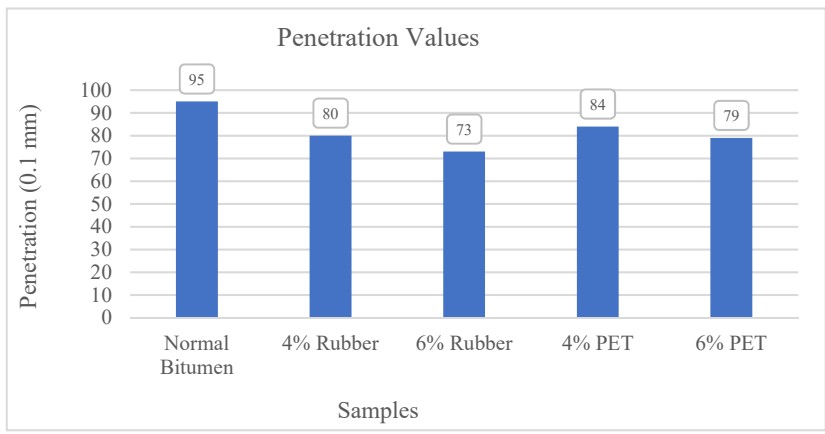

**Figure 2.** The values of penetration test of samples.

Furthermore, Figure 2 demonstrates that incorporating 4% and 6% of waste PET into bitumen reduced the penetration values by 11.5% and 16.8%, respectively, as shown in Figure 3. This behavior suggests that the availability of waste PET contributes to a pavement that is more resistant to deformation, such as rutting. This observation is supported by the findings of Kalantar et al. [29], who also found that adding waste PET to bitumen increases its consistency and hardness.

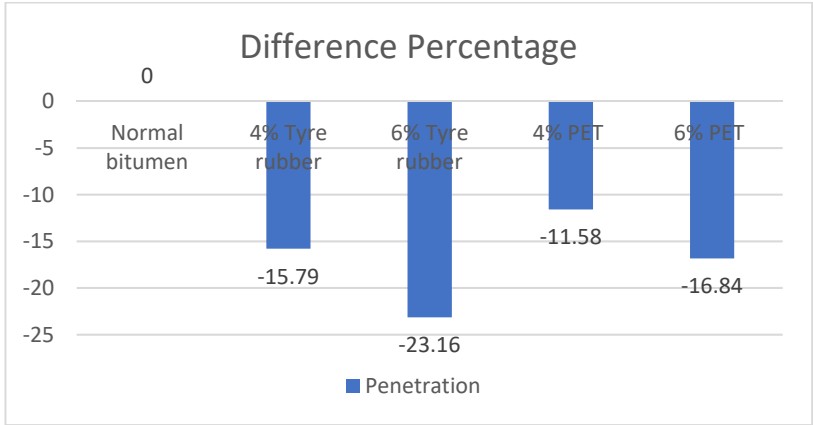

**Figure 3.** The differentiated rate of penetration values of modified samples to normal bitumen values.

In a study conducted by Manan et al. [30] involving plastic and crumb rubber, it was observed that the penetration value of plastic-modified bitumen decreased more compared to crumb rubber-modified bitumen.

Figure 4 displays the softening point values of different bitumen samples determined through a ring and ball test. The softening point represents the average temperature at which bitumen starts to soften, as discussed by Mashaan et al. [8]. They observed that the rubber content from tyres significantly influences the softening point values, in contrast to the size and type of tyre. They explained that this increase is attributed to the enhanced stiffness of the bitumen resulting from interactions between the tyre, rubber, and bitumen, leading to an increase in the molecular weight of the bitumen.

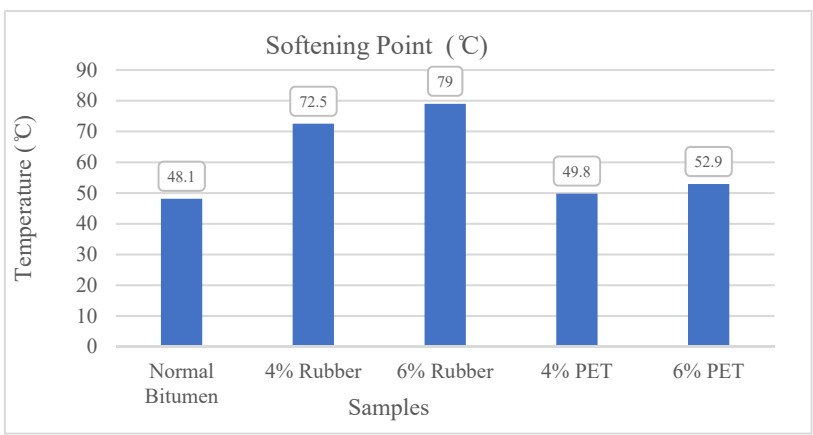

**Figure 4.** Softening point results of bitumen samples.

Figure 4 illustrates the impact of waste PET and tyre rubber content on the softening point of the bitumen sample. The utilization of 4% and 6% of waste tyre rubber improved the softening point by 50.73% and 64.24%, respectively, as shown in Figure 5. In addition, the incorporation of 4% and 6% of waste PET had a minor effect on the bitumen, resulting in slight increases in the softening points, reaching 49.8 °C and 52.8 °C, respectively, which is in agreement with the studies conducted by Kalantar et al. [29] and Suresh and Rao [31]. These studies investigated the effect of PET on various bitumen properties. They concluded that the addition of PET enhances the heat resistance of bitumen, making it less susceptible to temperature changes.

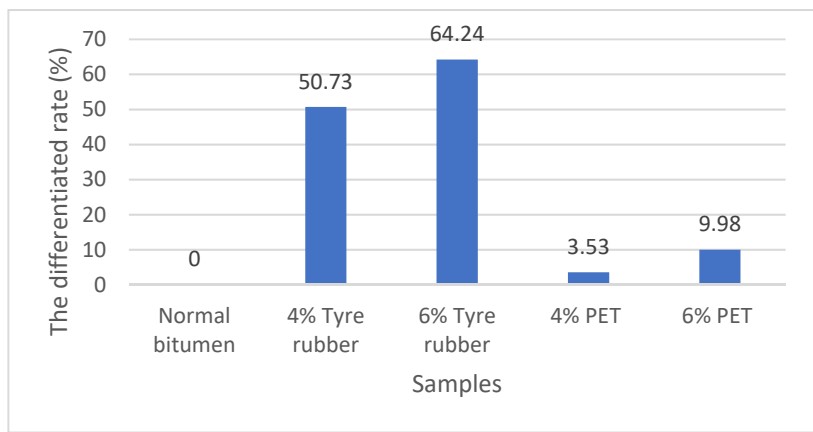

**Figure 5.** The differentiated rate of softening point values of modified samples to normal bitumen values.

However, it should be noted that some researchers have pointed out that PET is not suitable for bitumen production using the wet process due to its high melting point, which can approach 250 °C. In such cases, PET tends to separate from the bitumen, resulting in a non-uniform blend [32–34]. Ahmed and Mahdi [34] examined the effect of PET on bitumen properties using six different ratios ranging from 2% to 12%. They observed that increasing the PET content led to higher softening points.

Additionally, Manan et al. [30] reported that the use of PET in bitumen led to a more significant increase in the softening point compared to the use of crumb rubber.

Viscosity measures the resistance to the flow of a fluid, representing the internal friction within a moving fluid. In this study, the viscosity of bitumen was evaluated using the Saybolt Two-Tube Viscometer test. This test measures the flow rate of bitumen to assess its viscosity. A longer flow time indicates higher viscosity. Figure 6 displays normal and modified bitumen viscosity values and flow times. It is evident from Figure 6 that the flow time of normal bitumen is 1800 s. In contrast, the flow time for the tyre rubber-

modified bitumen samples is 2400 s and 2570 s for 4% and 6%, respectively. These increased flow times indicate that increased rubber content leads to higher bitumen viscosity. The increase ratio was 33% and 43% for the 4% and 6% tyre rubber-modified bitumen samples, respectively, as shown in Figure 7.

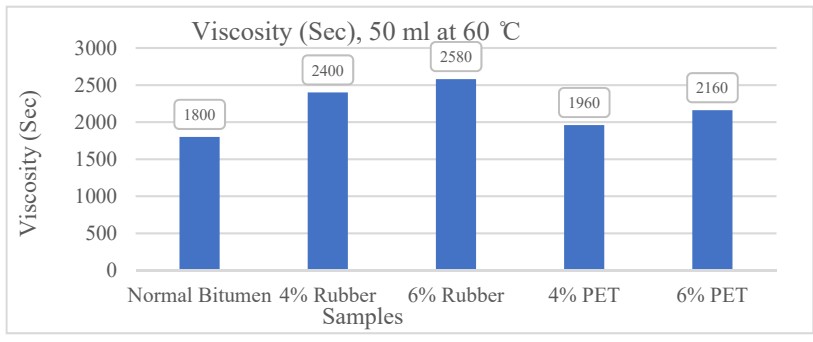

**Figure 6.** Viscosity results of normal and modified bitumen samples.

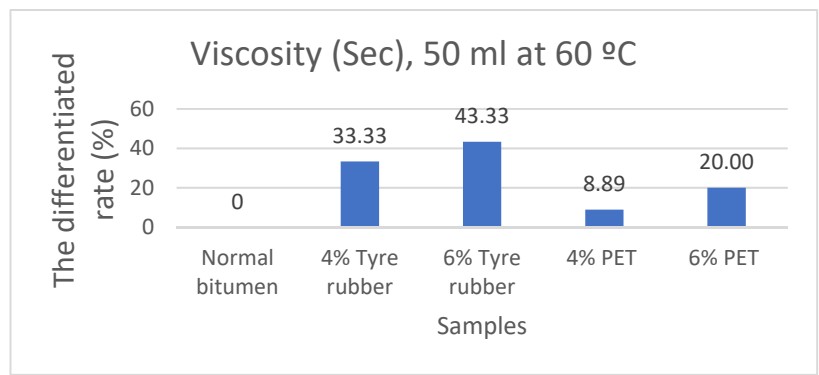

**Figure 7.** The differentiated rate of viscosity values of modified samples to normal bitumen values.

The flow time for the 4% PET-modified bitumen sample is 1960 s, while for the 6% PET-modified bitumen sample, it is 2160 s. The increase ratios were 8.9% and 20% for the 4% and 6% PET-modified bitumen samples, respectively, as depicted in Figure 7. During heating, tyre rubber was added to the bitumen, resulting in the interaction between the bitumen and tyre rubber. This interaction causes the rubber to swell, creating a gel-like material. The swelling reduces the distance between particles and increases the probability of particle collisions, thereby raising the viscosity of the bitumen [28–30]. Ahmad and Mahdi [34] demonstrated that the viscosity of bitumen increases with increasing percentages of PET added to the bitumen, up to 12%. However, beyond that point, the bitumen's viscosity remains relatively constant. Mahdi et al. [35] determined that incorporating polyethylene terephthalate (PET) in asphalt mixtures led to an increase in the softening temperature and a decrease in the penetration value, increasing the Performance Grade (PG) index. The PG index increased by 308% and 373% for 75 μm and 150 μm PET particle sizes, respectively, when the PET content was increased from 0% to 10%.

The experimental results obtained from both normal and modified bitumen samples generally demonstrate that the addition of waste tyre rubber has a more significant impact on normal bitumen than waste PET.

Manan et al. [30] conducted a study where they added tyre rubber and PET to bitumen. The findings indicated that PET had a more pronounced effect on bitumen than tyre rubber. They observed that the addition of PET resulted in a more significant decrease in the penetration value of bitumen compared to the addition of tyre rubber. Furthermore, the study demonstrated that the softening point of bitumen increased to a greater extent with PET than with tyre rubber.

High viscosity in bitumen at high temperatures is crucial for achieving excellent road paving results. This is because the viscosity of bitumen determines its ability to flow through an asphalt plant and coat aggregates in an asphalt concrete mix. In this study, the viscosity properties of asphalt binder were examined at two different temperatures: 135 °C and 165 °C. The results indicate that the modified binder with the addition of plastic waste exhibits higher viscosity at 135 °C than normal bitumen. As the percentage of plastic waste increases, the viscosity of the binder also increases, indicating a higher viscosity. The graph presented in the study shows that the viscosity of the original bitumen 60–70 at 135 °C is 500 cP, which is lower than that of the bitumen with 6% plastic waste addition [36–39].

In general, the results of the conducted tests confirm that waste tyre rubber has more beneficial effects on bitumen performance than PET waste. Studies by Manan et al. [30], Casey et al. [32], and Modarres and Hamedi [33] indicate that PET is not suitable for the wet process due to its high melting point, which can reach up to 250 °C.

Ben Zair et al. [40] reported that using PET waste with bitumen offers a suitable waste disposal solution and is beneficial for the environment, considering the large amount of PET waste generated.

Figure 8 illustrates the relationship between the penetration and viscosity results of all normal and modified bitumen samples. The results show that an increase in sample viscosity leads to a decrease in penetration values.

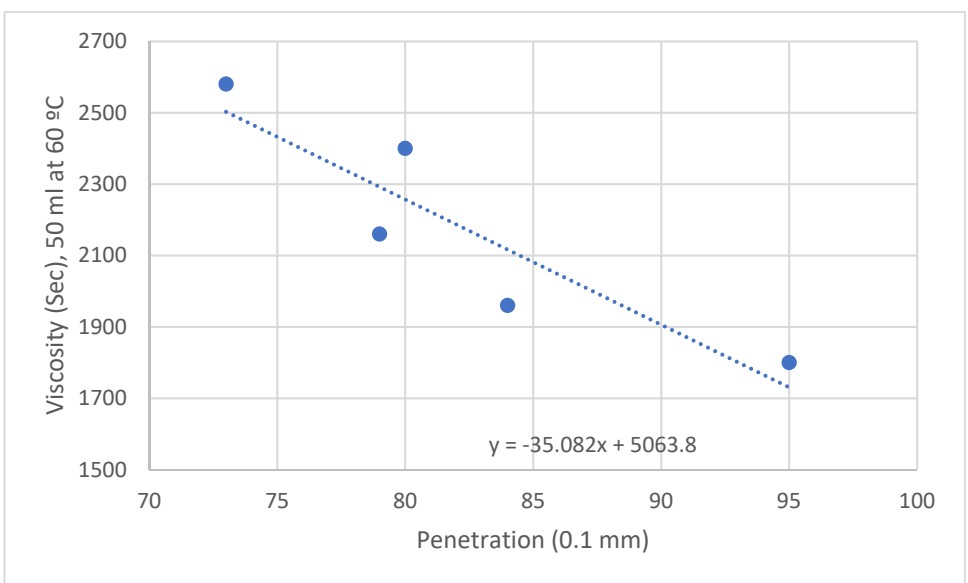

**Figure 8.** Relationship between data of penetration test and viscosity test.

## 4. Conclusions

The following conclusions can be drawn from this investigation:

1. The engineering properties of bitumen have been enhanced by incorporating waste PET and tyre rubber.
2. The utilization of waste PET and waste tyre rubber in bitumen resulted in improved penetration values (ranging from 73 to 80), softening points (ranging from 50 °C to 79 °C), and viscosity (ranging from 1960 to 2580 s).
3. The addition of waste PET and waste tyre rubber improved the consistency of bitumen.
4. Incorporating waste PET or waste tyre rubber into bitumen reduces its susceptibility to temperature changes.
5. Incorporating 6% waste PET or 6% waste tyre rubber into bitumen increases its viscosity by 20% and 43%, respectively.
6. The results indicate that waste tyre rubber has a more significant impact on bitumen than waste PET.

7. The viscosity and softening point of bitumen grade 85/100 in Oman can be enhanced by incorporating 6% waste rubber and 6% PET, especially in severe conditions with higher temperatures during summer weather.

8. Several studies have reported that adding crumb rubber and PET flakes enhances the softening point, consistency, and viscosity of bitumen.

9. Lastly, the use of crumb rubber waste and PET waste is considered environmentally friendly.

**Author Contributions:** Conceptualization, O.R.K.; methodology, O.R.K.; validation, M.M.F.; formal analysis, O.R.K. and M.M.F.; investigation, L.K.N.A.G.; resources, O.R.K.; writing—original draft preparation, L.K.N.A.G.; writing—review and editing, M.M.F.; visualization, L.K.N.A.G.; supervision, M.M.F.; project administration, L.K.N.A.G.; funding acquisition, O.R.K. All authors have read and agreed to the published version of the manuscript.

**Funding:** This research received no external funding.

**Institutional Review Board Statement:** Not applicable.

**Informed Consent Statement:** Not applicable.

**Data Availability Statement:** Not applicable.

**Acknowledgments:** The authors would like to thank the technicians of the Civil Engineering Department laboratory for their help in the experimental works. Our thanks are also extended to the United Rubber Industries (URI) Company and Octal Company for providing us with the materials used in this study.

**Conflicts of Interest:** The authors declare no conflict of interest.

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
