# Peer review of "Enhancing Bitumen Properties through the Utilization of Waste Polyethylene Terephthalate and Tyre Rubber"

_sustainability, doi:10.3390/su15129298_

Round 1

Reviewer 1 Report

1.      Rewrite the abstract

2.      Please include in the abstract some conclusions of the results.

3.      The paper needs careful language editing.

4.      Highlights the novelty and your research findings.

5.      Please reduce the length of the introduction by reducing the last two paragraphs of the introduction. (Line numbers 107 to 137)

6.      What is the conclusion from the literature review?

7.      Add the research gap at the end of section 1.

8.      Please check: The properties of bitumen. (Line number 148)

9.      Check Table 3. The last column may be standard not specification.

10.  Explain the sample preparation precisely.

11.  Rewrite the title of section 2.3.

12.  Please rewrite sections 2.3.1, 2.3.2, 2.3.3. As of now, it seems to be a laboratory manual. Please refer to other research articles to demonstrate the testing procedure.

13.  Thermometer (Ahmad, 2013) – What do you want to convey from this? Please rewrite it clearly.

14.  Thermometer (ASTMD 2019) – What do you want to convey from this? Please rewrite it clearly.

15.  Mention the correct standards used for tests.

16.  The tyre rubber used in this project was brought from United Rubber Industries (URI), and its size is between 1mm and 3 mm – Not clear. Please check.

17.  Don’t use in this project – Use in this paper or in this work.

18.  Add more properties in Table 3.

19.  Improve the quality of Figure 1.

20.  No need for Figure 2. Please remove it.

21.  Check the sample details in Table 4. The second sample may be Normal bitumen + 4% tyre rubber. Similarly, check other samples also.

22.  Check tables 5,6, and 7 also.

23.  Line number 330 (Both studies researched the effect of PET on some properties of bitumen.) – Rewrite.

24.  Please include either Table 5 or Figure 3. Table 6 or Figure 4. Table 7 or Figure 5.

25.  The definition of viscosity is resistance to flow (Line 351) – Please include the correct scientific definition with some references.

26.  “Jeong et al. [12] studied the effect of crumb rubber on bitumen viscosity. Bitumen had added two percentages of tire rubber, 10% and 20%. They determined that 20% of tire rubber gave a higher viscosity bitumen than 10%. Moreover, Khalid and Artamendi [29] indicated that the viscosity of bitumen increases by adding crumb rubber, with an increase of more than 10%” which may be included in the literature review.

27.  Rewrite the conclusions.

28.  What is the limitation of the work? What is the future scope of your research? Explain.

29.  The results obtained must be compared and contrasted with the literature data clearly.

30.  The paper should be completely rewritten. The authors may refer to some articles in this journal.

31. The title can be modified.

The paper needs careful language editing. 

Author Response

  1. Rewrite the abstract.

Answer: The abstract section is now updated; please refer to the revised manuscript.

  1. Please include in the abstract some conclusions of the results.

Answer: The abstract section is now updated; please refer to the revised manuscript.

  1. The paper needs careful language editing.

Answer: The paper English language is now professionally edited; please refer to the revised manuscript.

  1. Highlights the novelty and your research findings.

Answer: The paper is now updated to highlight the novelty of the research; please refer to the revised manuscript.

  1. Please reduce the length of the introduction by reducing the last two paragraphs of the introduction. (Line numbers 107 to 137)

Answer: The manuscript is now updated; however, as per the Editor's request, we cannot change the length of the manuscript significantly in order to keep the number of words within the acceptable limit; please refer to the revised manuscript.

  1. What is the conclusion from the literature review?

Answer: The manuscript is now updated, and the conclusion from the literature review is now added at the end of the introduction section; please refer to the revised manuscript.

  1. Add the research gap at the end of section 1.

Answer: The manuscript is now updated, and the research gap is added at the end of the introduction section; please refer to the revised manuscript.

  1. Please check: The properties of bitumen. (Line number 148)

Answer: The manuscript is now updated; please refer to the revised manuscript.

  1. Check Table 3. The last column may be standard not specification.

Answer: The manuscript is now updated; please refer to the revised manuscript.

  1. Explain the sample preparation precisely.

Answer: The manuscript is now updated; please refer to the revised manuscript.

  1. Rewrite the title of section 2.3.

Answer: The manuscript is now updated; please refer to the revised manuscript.

  1. Please rewrite sections 2.3.1, 2.3.2, 2.3.3. As of now, it seems to be a laboratory manual. Please refer to other research articles to demonstrate the testing procedure.

Answer: The manuscript is now updated; please refer to the revised manuscript.

  1. Thermometer (Ahmad, 2013) – What do you want to convey from this? Please rewrite it clearly.

Answer: This is now deleted, and the manuscript is now updated; please refer to the revised manuscript.

  1. Thermometer (ASTMD 2019) – What do you want to convey from this? Please rewrite it clearly.

Answer: This is now deleted, and the manuscript is now updated; please refer to the revised manuscript.

  1. Mention the correct standards used for tests.

Answer: The manuscript is now updated; please refer to the revised manuscript.

  1. The tyre rubber used in this project was brought from United Rubber Industries (URI), and its size is between 1mm and 3 mm – Not clear. Please check.

Answer: The manuscript is now updated; please refer to the revised manuscript.

  1. Don’t use in this project – Use in this paper or in this work.

Answer: The manuscript is now updated; please refer to the revised manuscript.

  1. Add more properties in Table 3.

Answer: The manuscript is now updated; please refer to the revised manuscript.

  1. Improve the quality of Figure 1.

Answer: The manuscript is now updated; please refer to the revised manuscript.

  1. No need for Figure 2. Please remove it.

Answer: The manuscript is now updated; please refer to the revised manuscript.

  1. Check the sample details in Table 4. The second sample may be Normal bitumen + 4% tyre rubber. Similarly, check other samples also.

Answer: The manuscript is now updated; please refer to the revised manuscript.

  1. Check tables 5,6, and 7 also.

Answer: The manuscript is now updated; please refer to the revised manuscript.

  1. Line number 330 (Both studies researched the effect of PET on some properties of bitumen.) – Rewrite.

Answer: The manuscript is now updated; please refer to the revised manuscript.

  1. Please include either Table 5 or Figure 3. Table 6 or Figure 4. Table 7 or Figure 5.

Answer: The manuscript is now updated; please refer to the revised manuscript.

  1. The definition of viscosity is resistance to flow (Line 351) – Please include the correct scientific definition with some references.

Answer: The manuscript is now updated; please refer to the revised manuscript.

  1. “Jeong et al. [12] studied the effect of crumb rubber on bitumen viscosity. Bitumen had added two percentages of tire rubber, 10% and 20%. They determined that 20% of tire rubber gave a higher viscosity bitumen than 10%. Moreover, Khalid and Artamendi [29] indicated that the viscosity of bitumen increases by adding crumb rubber, with an increase of more than 10%” which may be included in the literature review.

Answer: The manuscript is now updated; please refer to the revised manuscript.

  1. Rewrite the conclusions.

Answer: The manuscript is now updated; please refer to the revised manuscript.

  1. What is the limitation of the work? What is the future scope of your research? Explain.

Answer: The manuscript is now updated; please refer to the revised manuscript.

  1. The results obtained must be compared and contrasted with the literature data clearly.

Answer: The manuscript is now updated; please refer to the revised manuscript.

  1. The paper should be completely rewritten.

Answer: The manuscript is now updated; please refer to the revised manuscript.

  1. The title can be modified.

Answer: The manuscript title is now updated; please refer to the revised manuscript.

Reviewer 2 Report

The industrialization of modern society has produced and consumed extremely large quantities of plastic and rubber products. The waste generated by the use of these products is usually a hazardous waste that cannot be degraded by nature, especially discarded tires and plastics for daily use. In this study, the authors used rubber and polyethylene terephthalate (PET), typical recyclables from waste tires and plastics, as binders to improve the properties of asphalt materials, and found that this polymer material had a significant improvement in the performance of asphalt. This work maximizes the recycling efficiency of plastic and rubber products and contributes to the optimization of road pavements. However, the analysis of this work is too rough. In addition, there are too many mistakes for this manuscript. Before the article can be officially published, not only the first draft needs to be refined, but also some additional performance testing work needs to be added.

1.    For asphalt applied in road construction, a series of tests on modified friction coefficients are also required to ensure the practical potential of asphalt with rubber or PET as filler.

2.    Details of material preparation and performance testing in sections 2.2 and 2.3, as well as equipment information, should be collated in the supporting information.

3.    The conclusion section is too concise and should emphasize the value of asphalt modified by rubber or PET in practical applications.

4.    The formatting in many tables in the text is not appropriate, e.g., the data in Table 1 are not aligned.

5.    1. The figure notes in Figure 1 are not aligned with the images. There are many similar formatting problems in the text, which need to be checked carefully.

6.     There is too much blank space between some subsections, such as before 2.3.2 and before 2.3.3.

English very difficult to understand/incomprehensible

Author Response

  1. For asphalt applied in road construction, a series of tests on modified friction coefficients are also required to ensure the practical potential of asphalt with rubber or PET as filler.

Answer: That is correct; however, the scope of this study is limited to the tests mentioned in the manuscript, and these extra tests can be the subject of future studies.  

  1. Details of material preparation and performance testing in sections 2.2 and 2.3, as well as equipment information, should be collated in the supporting information.

Answer: The manuscript is now updated; please refer to the revised manuscript.

  1. The conclusion section is too concise and should emphasize the value of asphalt modified by rubber or PET in practical applications.

Answer: The manuscript is now updated; please refer to the revised manuscript.

  1. The formatting in many tables in the text is not appropriate, e.g., the data in Table 1 are not aligned.

Answer: The manuscript is now updated; please refer to the revised manuscript.

  1. The figure notes in Figure 1 are not aligned with the images. There are many similar formatting problems in the text, which need to be checked carefully.

Answer: The manuscript is now updated; please refer to the revised manuscript.

  1. There is too much blank space between some subsections, such as before 2.3.2 and before 2.3.3.

Answer: The manuscript is now updated; please refer to the revised manuscript.

Round 2

Reviewer 1 Report

Congratulations. The quality of the paper is improved now. 

Reviewer 2 Report

I suggest to accept the manuscript in the current form.